# EXPLORING IMAGE GENERATION VIA MUTUALLY EX-CLUSIVE PROBABILITY SPACES AND LOCAL DEPENDENCE HYPOTHESIS

## ABSTRACT

A common assumption in probabilistic generative models for image generation is that learning the global data distribution suffices to generate novel images via sampling. We investigate the limitation of this core assumption, namely that learning global distributions leads to memorization rather than generative behavior. We propose two theoretical frameworks, the Mutually Exclusive Probability Space (MEPS) and the Local Dependence Hypothesis (LDH), for investigation. MEPS arises from the observation that deterministic mappings (e.g., neural networks) involving random variables tend to reduce overlap coefficients among involved random variables, thereby inducing exclusivity. We further propose a lower bound in terms of the overlap coefficient, and introduce a Binary Latent Autoencoder (BL-AE) that encodes images into signed binary latent representations. LDH formalizes dependence within a finite observation radius, which motivates our $\gamma$-Autoregressive Random Variable Model ($\gamma$-ARVM). $\gamma$-ARVM is an autoregressive model, with a variable observation range $\gamma$, that predicts a histogram for the next token. Using $\gamma$-ARVM, we observe that as the observation range increases, autoregressive models progressively shift toward memorization. In the limit of global dependence, the model behaves as a pure memorizer when operating on the binary latents produced by our BL-AE. Comprehensive experiments and discussions support our investigation.

## 1 INTRODUCTION

Probabilistic generative models, such as Variational Autoencoders (VAEs), Generative Adversarial Networks (GANs), diffusion models, and autoregressive models have achieved remarkable progress in image generation. A core assumption is that these models, learn an image distribution from which new images can be generated via sampling (Bond-Taylor et al., 2022). However, we explore a potential limitation of this assumption; namely, that learning global distributions[1] results in memorization rather than generative behavior. Specifically, we focus on autoregressive models. For this investigation, we introduce two theoretical frameworks. The first,

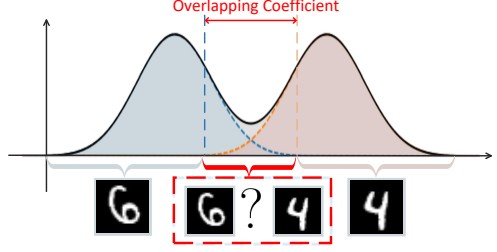

Figure 1: Selecting images for values in the overlap range is ambiguous.

Mutually Exclusive Probability Space (MEPS), arises from the observation that deterministic mappings involving random variables tend to reduce the overlap coefficients inherent in the system. This reduction makes the probability spaces of the random variables effectively mutually exclusive. The second is the Local Dependence Hypothesis (LDH), which is motivated by an analysis of why autoregressive models tend to reproduce training samples. While this phenomenon is often attributed to overfitting, we argue that it is related to the core assumption of learning global distributions. The

---

[1]By global distribution we mean the overall probability distribution that the generative model is trained to approximate across the entire dataset.

issue lies in differing philosophical views between the frequentist and Bayesian interpretations of whether probability distributions objectively exist. This leads us to propose the Local Dependence Hypothesis (LDH), which posits that generative capacity in autoregressive models arises from modeling local dependence rather than global distributions.

In a trainable deterministic mapping from random variables to deterministic variables, for example, a network that takes noise as input for image reconstruction (like VAEs, GANs, or diffusion models), the distributions of the random variables may overlap. In such cases, observations from different optimization steps within the overlap region may be optimized toward inconsistent targets. This is especially true when training for many epochs. Consequently, such inconsistent optimization targets raise the lower bound of the entire mapping system (Theorem 3.3), thereby degrading mapping fidelity (specifically, reconstruction quality). As shown in Fig. 1, observations from overlapping ranges confuse the final optimization target. When the random variables are also parameterized for optimization, the learning dynamics tend to diminish such overlapping ranges, and the means of these random variables are pushed apart (Theorem 3.5). Exclusivity thus emerges. This observation motivates the formulation of the Mutually Exclusive Probability Space (MEPS) (Definition 3.1). Leveraging this exclusivity, we propose the Binary Latent Autoencoder (BL-AE), which encodes images into binary latent representations. However, when feeding the learned binary latents into PixelCNN (van den Oord et al., 2016), a widely used autoregressive model, the network often reproduces training samples. This motivates our concern that learning global distributions leads to memorization. To investigate this possibility, we propose the Local Dependence Hypothesis (LDH), which is formalized by assuming a bounded dependence radius for autoregressive models (Assumption 4.1). Based on LDH, the $\gamma$-Autoregressive Random Variable Model ($\gamma$-ARVM) is proposed, which is an autoregressive model with a variable observation range $\gamma$. In addition, given the subtle presence of MEPS in autoregressive models (Sec. 4.2), the proposed $\gamma$-ARVM outputs histograms describing the distribution of the next token rather than a label like PixelCNN. The main contributions of this work are:

- We propose the Mutually Exclusive Probability Space (Definition 3.1) by observing exclusivity in an optimizable deterministic mapping system from random variables to deterministic targets. Based on this exclusivity, the Binary Latent Autoencoder (BL-AE) is introduced. In particular, by injecting noise into the outputs of activation functions with limited support width, the model learns signed binary latents, which are naturally used as tokens for autoregressive models. Moreover, MEPS can also be applied to revise the priors of generative models such as VAEs (Sec. A.1.1) for improving fidelity.

- We propose the Local Dependence Hypothesis (LDH) (Assumption 4.1) to investigate a potential limitation in the core assumption of probabilistic generative models; namely, that learning global latent distributions may lead to memorization rather than generative behavior. In particular, the $\gamma$-Autoregressive Random Variable Model is proposed. Unlike previous autoregressive models that typically imply global dependence, the proposed $\gamma$-ARVM has a variable observation range $\gamma$. Using $\gamma$-ARVM, we observe that as the observation range increases, autoregressive models progressively shift toward memorization (Sec. 5.2).

## 2  RELATED WORK

Probabilistic generative models have achieved remarkable progress across a range of applications. A core assumption is that models learn a data distribution from which new content can be generated via sampling (Bond-Taylor et al., 2022). For example, Variational Autoencoders (VAEs) (Kingma & Welling, 2014) assume a Gaussian prior over latent variables and maximize the evidence lower bound (ELBO) to approximate the true posterior. Generative Adversarial Networks (GANs) (Goodfellow et al., 2014) employ an adversarial objective, wherein a generator and a discriminator are trained in opposition. Despite ongoing debate regarding whether GANs learn the true data distribution (Arora et al., 2018; Chen et al., 2022), empirical results demonstrate the effectiveness of GANs in image generation (Lee et al., 2025). Diffusion models (Ho et al., 2020b), or score-based models (Song et al., 2020), learn to generate data by reversing a diffusion process through score-function estimation. Autoregressive models (Chen & Pan, 2025; Cheng et al., 2025) factorize the joint distribution into a product of conditionals and are usually combined with discrete latent quantization methods such as VQ-VAE (van den Oord et al., 2017). There are also other generative models such

as energy-based models (Gao et al., 2021) and normalizing flows (Tabak & Turner, 2013; Papamakarios et al., 2019; Stimper et al., 2022; Vuckovic). Most of these models share a fundamental assumption that learning a global distribution—whether over data or latent representations—is often traced back to the manifold hypothesis (Bengio et al., 2013). In this work, we propose the Mutually Exclusive Probability Spaces (MEPS) and the Local Dependence Hypothesis (LDH) to explore a potential limitation of this assumption.

Although our MEPS framework is newly proposed, the underlying principle can be observed in several previous works. For example, the inconsistent optimization target is related to the optimization inconsistency in $\beta$-VAE (Higgins et al., 2017), where the weights of the KL loss and the reconstruction loss are controlled by user-defined parameters. Burgess et al. (Burgess et al., 2018) explain this inconsistency through the information bottleneck, while Lucas et al. (Lucas et al., 2019) suggest that it leads to posterior collapse. Recent work (Michlo et al., 2023) has also discussed the disentanglement of the reconstruction loss. Moreover, the inconsistency can also be observed in the "prior hole" problem (Aneja et al., 2021; Xiao et al., 2020; Nalisnick et al., 2018), considering that a single Gaussian prior is insufficient for modeling complex data distributions (Vahdat & Kautz, 2020). In contrast, Gaussian Mixture VAEs (GMVAEs) (Dilokthanakul et al., 2016; Yang et al., 2019; Guo et al., 2020) replace the standard prior with a mixture of Gaussians, which reduces the overlap between the distributions of different latent variables, thereby alleviating the optimization inconsistency. In this work, we mathematically demonstrate the exclusivity of these probability spaces and propose the Binary Latent Autoencoder (BL-AE).

The Local Dependence Hypothesis (LDH) can be viewed as an extension or improvement of the global dependence implicitly assumed in most autoregressive models (van den Oord et al., 2016). Typically, autoregressive models imply global dependence, since they factorize the joint distribution into full-context conditionals. However, there are also autoregressive models that incorporate local patterns (Mao et al., 2024; Cao et al., 2021), most of which were proposed primarily to reduce computational complexity. For example, Cao et al. (Cao et al., 2021) proposed a Local Autoregressive Transformer that restricts attention regions to accelerate inference. In contrast, our work is, to the best of our knowledge, the first to systematically argue that learning the global distribution can lead to memorization. Unlike prior work that devises attack methods to extract training samples from pre-trained large models such as Stable Diffusion (Ross et al., 2025; van den Burg & Williams, 2021; Kowalczuk et al., 2025; Kasliwal et al., 2025; Yu et al., 2025), our LDH serves as a theoretical framework to examine this foundational assumption in autoregressive models.

## 3 MUTUALLY EXCLUSIVE PROBABILITY SPACES

### 3.1 THEORETICAL FOUNDATIONS

**Definition 3.1 [Mutually Exclusive Probability Space (MEPS)].** Let $\tilde{\mathbf{Z}} = \{\tilde{\mathbf{z}}_i\}_{i=1}^N$ be a set of random variables with densities $\{p_{\tilde{\mathbf{z}}_i}(\mathbf{z})\}_{i=1}^N$. Let $\mathbf{X} = \{\mathbf{x}_i\}_{i=1}^M$ be a set of deterministic variables with $M \leq N$. For each pair $(i, j)$, the overlap coefficient between $\tilde{\mathbf{z}}_i$ and $\tilde{\mathbf{z}}_j$ is:

$$\mathrm{OC}(\tilde{\mathbf{z}}_i, \tilde{\mathbf{z}}_j) = \int \min\big(p_{\tilde{\mathbf{z}}_i}(\mathbf{z}),\, p_{\tilde{\mathbf{z}}_j}(\mathbf{z})\big)\, d\mathbf{z}. \tag{1}$$

Let $d_\phi : \tilde{\mathbf{Z}} \to \mathbf{X}$ be a deterministic mapping. We say that $(\tilde{\mathbf{Z}}, d_\phi)$ forms a Mutually Exclusive Probability Space (MEPS) if:

$$\max_{i \neq j}\, \mathrm{OC}(\tilde{\mathbf{z}}_i, \tilde{\mathbf{z}}_j)\, \leq\, \varepsilon. \tag{2}$$

When $\varepsilon = 0$, we obtain a strict MEPS (all pairwise overlaps vanish up to measure zero). When $\varepsilon > 0$ is small, we obtain an approximate MEPS (pairwise overlaps are reduced to a negligible measure). Note that strict MEPS is rare, unless otherwise stated, "MEPS" in this paper refers to the approximate case.

**Remark 3.2.** The MEPS definition can be understood as a characterization of overlap coefficients among random variables under a deterministic mapping, such as a neural network decoder. It also reflects a training objective: learning dynamics tend to reduce pairwise overlaps, thereby pushing the latent space closer to a strict MEPS. Thus, the definition serves both as a descriptive criterion and as a motivation for optimization. In addition, the densities $\{p_{\tilde{\mathbf{z}}_i}(\mathbf{z})\}_{i=1}^N$ must be parameterized by optimizable parameters. Otherwise, overlaps remain fixed and cannot diminish. Note that MEPS

still exist in this case, but only in a fixed form. The deterministic mapping $d_\phi$ may be either trainable or fixed.

**Theorem 3.3 [Reconstruction MSE Lower Bound].** Let $\tilde{\mathbf{Z}} = \{\tilde{\mathbf{z}}_i\}_{i=1}^N$ be random variables with densities $\{p_{\tilde{\mathbf{z}}_i}\}_{i=1}^N$. In particular, each random variable is obtained by injecting additive noise, i.e., $\tilde{\mathbf{z}}_i = \mathbf{z}_i + \epsilon$, where for each $i$ the noise $\epsilon$ is drawn independently from the same unimodal, symmetric distribution. Let $\mathbf{X} = \{\mathbf{x}_i\}_{i=1}^N$ be deterministic targets. Suppose $d_\phi : \tilde{\mathbf{Z}} \to \mathbf{X}$ is a deterministic mapping (e.g., a neural decoder). For each $i$, we reconstruct $\mathbf{x}_i$ as $d_\phi(\tilde{\mathbf{z}}_i)$ and evaluate the reconstruction error using the mean squared error (MSE). Then the mean reconstruction loss satisfies:

$$\frac{1}{N}\sum_{i=1}^N \mathbb{E}_\epsilon\left[\|d_\phi(\tilde{\mathbf{z}}_i) - \mathbf{x}_i\|^2\right] \ \geq \ \frac{1}{4N^2}\sum_{i,j=1}^N \mathrm{OC}(\tilde{\mathbf{z}}_i, \tilde{\mathbf{z}}_j)\,\|\mathbf{x}_i - \mathbf{x}_j\|^2. \tag{3}$$

The proof is provided in Sec. A.2.1.

**Remark 3.4.** This lower bound implies that the average reconstruction MSE cannot approach zero whenever the pairwise overlaps are nonzero. Consequently, the reconstructed images cannot perfectly match the training targets under nonzero overlap. In addition, although this lower bound is derived under the mean squared error, a similar bound can be established for other convex loss functions. This is because the constant term on the right-hand side arises from the convexity inequality and is independent of the overlap coefficient.

**Theorem 3.5 [Mutual Exclusivity Theorem].** Let $p_{\tilde{\mathbf{z}}_i}$ and $p_{\tilde{\mathbf{z}}_j}$ be unimodal and symmetric densities centered at means $\mathbf{z}_i, \mathbf{z}_j \in \mathbb{R}^d$. Then minimizing the expectation of overlap coefficient satisfies:

$$\operatorname*{argmin}_{\mathbf{z}_i, \mathbf{z}_j} \mathbb{E}_\epsilon[\mathrm{OC}(\tilde{\mathbf{z}}_i, \tilde{\mathbf{z}}_j)] \Rightarrow \operatorname*{argmin}_{\mathbf{z}_i, \mathbf{z}_j} \mathbb{E}_\epsilon[\mathrm{OC}(\mathbf{z}_i + \epsilon, \mathbf{z}_j + \epsilon)] \Rightarrow \operatorname*{argmax}_{\mathbf{z}_i, \mathbf{z}_j} \frac{1}{N^2}\sum_{i,j}\|\mathbf{z}_i - \mathbf{z}_j\|^2. \tag{4}$$

The proof is provided in Sec. A.2.2

**Remark 3.6.** Reducing pairwise overlaps under training dynamics forces the random variables to separate in expectation, thereby encouraging the formation of mutual exclusivity between random variables whose distributions exhibit overlap.

**MEPS in VAEs, diffusion models and GANs:**
MEPS exist in a wide range of probabilistic generative models, including VAEs, diffusion models, GANs, and even autoregressive models. The presence of MEPS implies that reconstructed images cannot perfectly match training samples, which effectively prevents overfitting. However, this also degrades reconstruction fidelity and thus leads to lower generation quality. Since probabilistic generative models aim to approximate the data distribution, the trade-off between memorization and generalization becomes critical. The overlap coefficient (OC), which characterizes MEPS, provides a natural measure of this trade-off. As shown in Fig. 2, diffusion models apply a fixed noise schedule, resulting in a fixed OC. VAEs involve a competition between the KL term and the reconstruction loss, yielding a variable OC (typically less than 1). In contrast, GANs sample directly from noise without explicit latent constraints, effectively corresponding to an OC of 1 (Sec. A.2.3). As a result, diffusion models tend to be the easiest to train,

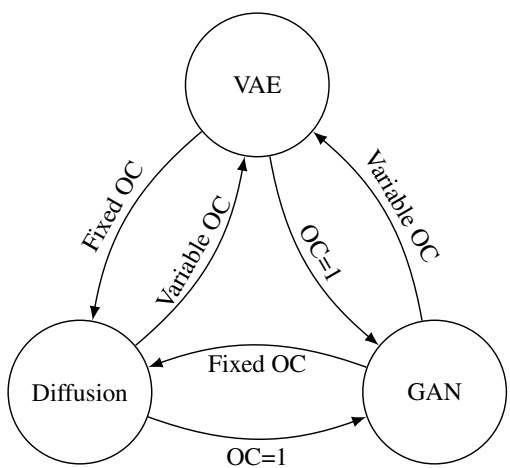

Figure 2: VAE, GAN, and diffusion models are correlated by different assumptions introducing the overlap coefficient (OC) in the mutually exclusive probability space.

while GANs are generally the most difficult. Moreover, according to our Theorem 3.3, a lower overlap coefficient in the Mutually Exclusive Probability Space leads to lower reconstruction quality, which in turn typically results in better FID scores, due to the memorization . Based on this observation, the fidelity of generated images (a better FID) can be improved by choosing priors that induce a lower overlap coefficient. For example, in VAEs, a Gaussian prior with a smaller variance

$\sigma$ achieves better FID values (Sec. A.1.1). While such behavior may reduce diversity, generative modeling is essentially a trade-off between fidelity and diversity. Furthermore, by revising the prior assumptions of VAEs (Sec. A.1.2) , GANs (Sec. A.1.3), and diffusion models (Sec. A.1.4), one can deliberately drive these models toward memorization behavior, which highlights the significance of MEPS in studying the memorization properties of generative models.

## 3.2 BINARY LATENT AUTOENCODER

The Binary Latent Autoencoder is a practical application leveraging the exclusivity in MEPS. We employ an activation function with a bounded output range, such as the hyperbolic tangent (tanh). Then, noise from a symmetric bounded distribution is injected into the activation function's output, thereby extending the output into random variables, with the motivation to form MEPS. Due to exclusivity, the network implicitly pushes these latent variables toward distinct, non-overlapping regions at the limits of the tanh activation ($\pm 1$). As training proceeds[2], the activation outputs converge to binary values $\{-1, 1\}$, resulting in an autoencoder with discrete signed binary latents. Mathematically:

$$\text{BL-AE}(\mathbf{x}_i; \theta, \phi) = \sum_{\mathbf{x}_i \in \mathbf{X}} \|d_\phi(\psi(e_\theta(\mathbf{x}_i)) + \sigma \cdot \epsilon) - \mathbf{x}_i\|^2, \tag{5}$$

where $\psi(\cdot)$ is the activation function. $d_\phi, e_\theta$ denote the decoder and the encoder with $\phi$ and $\theta$ as their parameters. $\mathbf{x}_i$ denotes an image from dataset $\mathbf{X}$. $\epsilon$ is a noise following a distribution with unimodal and symmetric densities, such the Gaussian distribution, or the generalized triangular distribution (GTD):

$$\epsilon \sim \text{Tri}(\kappa) = \begin{cases} (1 - u^\kappa), & \text{if } u > 0^+ \\ (|u|^\kappa - 1), & \text{if } u < 0^-, \end{cases} \tag{6}$$

where $u \sim \mathcal{U}(-1, 1)$ is the uniform distribution from -1 to 1. The parameter $\kappa$ controls the sharpness of the distribution. The proposed BL-AE works well to learn quantization tokens. This is naturally suitable for autoregressive models. Thus, we input the tokens from BL-AE into autoregression, and the memorization appears, which motivated us to propose the local dependence hypothesis for further investigation.

## 4 LOCAL DEPENDENCE HYPOTHESIS

### 4.1 THEORETICAL FOUNDATIONS

**Assumption 4.1 [$\gamma$-Local Dependence Assumption ($\gamma$-LDA)].** Let $\{\tilde{\mathbf{z}}_i\}_{i=1}^N$ be random variables. Fix a radius parameter $\gamma > 0$ under a given distance metric $d(\cdot, \cdot)$. We assume that the mutual information between variables is bounded by a tolerance $\varepsilon$:

$$d(\tilde{\mathbf{z}}_i, \tilde{\mathbf{z}}_j) > \gamma \ \Rightarrow \ I(\tilde{\mathbf{z}}_i; \tilde{\mathbf{z}}_j) \leq \varepsilon. \tag{7}$$

Thus, $\gamma$ defines a bounded dependence radius. Beyond this radius, dependencies vanish up to $\varepsilon$-tolerance, while within it, dependencies can be arbitrary. When $\varepsilon = 0$, we obtain a strict LDA, where exact independence holds outside radius $\gamma$. When $\varepsilon > 0$ is small, we obtain an approximate LDA, where long-range dependencies are reduced to negligible levels.

**Remark 4.2.** The $\gamma$-LDA hypothesis is conceptually related to the n-gram assumption in language modeling, as both impose locality by restricting the range of dependencies. The n-gram assumption relies on a fixed-size window to truncate dependencies, primarily for natural language sequences. In contrast, $\gamma$-LDA controls locality through mutual information with tolerance, making it applicable to more general settings such as images and other high-dimensional data. Therefore, $\gamma$-LDA can be regarded as a generalization of the n-gram assumption. Our $\gamma$-LDA is used to generalize autoregressive models. When the radius parameter $\gamma$ is greater than or equal to the sequence length, it reduces to standard autoregressive models such as PixelCNN (van den Oord et al., 2016). When the radius parameter $\gamma$ is smaller than the sequence length, it effectively yields a local autoregressive model. Thus, a variable observation range autoregressive model can be written as:

$$p(\mathbf{Z}) = \prod_{i=1}^N p(\mathbf{z}_i \mid \mathbf{z}_{<i}) \ \Rightarrow \ p(\mathbf{Z}) = \prod_{i=1}^N p(\mathbf{z}_i \mid \mathbf{z}_{[i-\gamma, i)}). \tag{8}$$

---

[2]Adding noise to the latent variables remains differentiable via the reparameterization trick, which allows gradients to pass through the stochastic sampling process.

## 4.2 $\gamma$-AUTOREGRESSIVE RANDOM VARIABLE MODEL

Unlike previous common autoregressive models in image generation that imply global dependence, the $\gamma$-Autoregressive Random Variable Model ($\gamma$-ARVM) is based on our $\gamma$-LDA, with variable observation ranges. Moreover, the output of the proposed $\gamma$-ARVM also differs from regular masked architectures such as PixelCNN or Transformer,[3] where both the input and the output are sequences of tokens. In our $\gamma$-ARVM, the input is token sequences within an observation range $\gamma$. The output of the proposed $\gamma$-ARVM is a histogram that describes the distribution of the next token. Hence, before training, the token distribution conditioned on the observation range $\gamma$ is captured in a statistical manner.

$$q(\mathbf{Z}) = \prod_{i=1}^{N} q(\mathbf{z}_i \mid \mathbf{z}_{[i-\gamma,i)}) = \prod_{i=1}^{N} \mathbb{P}(\mathbf{z}_i = \mathbf{k} \mid \mathbf{z}_{[i-\gamma,i)} = \mathbf{g}) = \prod_{i=1}^{N} \frac{\sum_{n=1}^{N} \mathbb{1}[\mathbf{z}_i^{(n)} = \mathbf{k}] \cdot \mathbb{1}[\mathbf{z}_{[i-\gamma,i)}^{(n)} = \mathbf{g}]}{\sum_{n=1}^{N} \mathbb{1}[\mathbf{z}_{[i-\gamma,i)}^{(n)} = \mathbf{g}]},$$
(9)

where $\mathbb{1}(\cdot)$ is the indicator function. $\mathbf{k}$ and $\mathbf{g}$ are specific values of output token and input token sequences. After capturing the training instances, the KL-divergence is used as the loss function:

$$\mathcal{L}(p(\mathbf{Z}), q(\mathbf{Z})) = \sum_{i=1}^{N} \int q(\mathbf{z}|\mathbf{z}_{[i-\gamma,i)}) \log \frac{q(\mathbf{z}|\mathbf{z}_{[i-\gamma,i)})}{p(\mathbf{z}|\mathbf{z}_{[i-\gamma,i)})} d\mathbf{z}.$$
(10)

**MEPS in Autoregressive Model:** The main reason we do not follow the common sequence-to-sequence training paradigm is to handle the subtle MEPS in previous autoregressive models. From the mathematical definition of autoregression in the left part Eq. 8, MEPS does not appear to exist. However, in practical implementations, MEPS inevitably emerges. The same input sequence may map to different next tokens, creating an inconsistent target that induces a non-vanishing lower bound for the cross-entropy loss. For example, consider the two binary sequences $(0, 0, 1, 0)$ and $(0, 0, 0, 0)$. From the first sequence, we obtain the mapping $(0, 0) \rightarrow 1$, while from the second we obtain $(0, 0) \rightarrow 0$. This condition forces an overlap that cannot vanish, preventing the loss from approaching zero. Our proposed ARVM addresses this issue by predicting a histogram of the output label, e.g., $(0, 0) \rightarrow q(y \mid (0, 0)) = [p_0, p_1] = [0.5, 0.5]$. In this way, the $\gamma$-ARVM can reduce the loss to extremely small values (e.g., $10^{-6}$), much smaller than those of sequence-to-sequence models such as PixelCNN. As a result, we are able to observe pure memorization conditions, which supports our claim that learning the global distribution tends to lead to memorization rather than genuine generative behavior. While this phenomenon is frequently described as overfitting, one may argue that, in strict logical terms, the concept is somewhat redundant. This is because a near-zero loss naturally signifies optimization with respect to the chosen objective (Sec. A.1.5).

## 5 EXPERIMENTS

### 5.1 MUTUALLY EXCLUSIVE PROBABILITY SPACE

#### 5.1.1 MUTUAL EXCLUSIVITY THEOREM

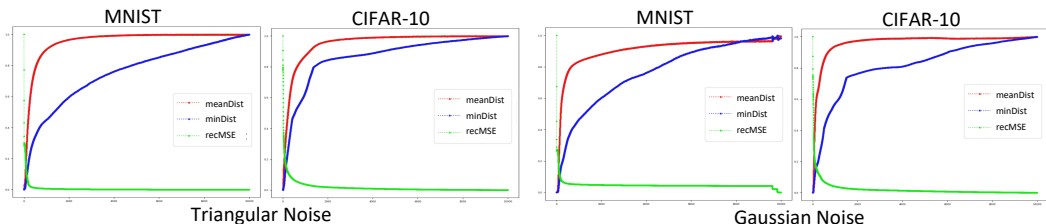

Figure 3: Demonstration of the mutual exclusivity theorem on the MNIST and CIFAR-10 datasets with Gaussian and triangular noise.

We demonstrate the exclusivity in MEPS on MNIST and CIFAR-10 datasets with Gaussian and triangular noise settings, to support the generalization of MEPS. In practice, the computation of overlap

---

[3]Both rely on masking to enable parallel training in autoregressive models.

coefficient (OC) in high dimensions is extremely expensive. Thus, we adopt an indirect approach by scaling the variance parameter $\sigma$. Increasing the value of $\sigma$ consistently increases the overlap for Gaussian and triangular distributions. In particular, a straightforward autoencoder for image reconstruction with the architecture described in Tab. 4 is utilized. During training, we inject noise into the latent variables $z_i$ to create random variables $\tilde{z}_i$ following different distributions (Gaussian and Triangular). Since mean squared error is used for reconstruction, the decoder becomes a deterministic mapping from latent random variables to deterministic targets (the ground-truth images). MEPS thus emerges, and these latent random variables become mutually exclusive. For demonstration, the MSE loss, the minimum distance between pairs of latents, and the average distance between the means of latent pairs are plotted in Fig. 3. Note that all curves are min–max normalized to $[0, 1]$ for visualization, with normalization details provided in Tab. 6. It is clear that the average distance between the means $z_i$ and $z_j$ of the latent variables $\tilde{z}_i$ and $\tilde{z}_j$ increases as the MSE decreases with an increasing number of training epochs. This behavior is consistent with the Mutual Exclusivity Theorem (Theorem 3.5).

### 5.1.2 RECONSTRUCTION MSE LOWER BOUND THEOREM

To further demonstrate the Reconstruction MSE lower bound, we fix the parameters of encoders in the previous section (Sec. 5.1.1), and increase the intensity of noise for ablation experiments. Specifically, we multiply the noise by a $\sigma$ value of $[0.5, 1.0, 1.5, 2.0, 2.5, 3.0]$, and then retrain the decoder with a sufficient number of epochs. When the value of $\sigma$ increases, the overlap coefficient increases as well. Then based on Theorem 3.3, the lower bound of the reconstruction error increases and leads to a decrease in reconstruction quality. Thus, the plot of $\sigma$ and average reconstruction quality evaluated by Peak Signa-to-Noise Ratio (PSNR) is demonstrated in Fig. 4. We can clearly observe the monotonic trend as $\sigma$ increases, the reconstruction quality decreases, which is consistent with the Reconstruction MSE Lower Bound Theorem.

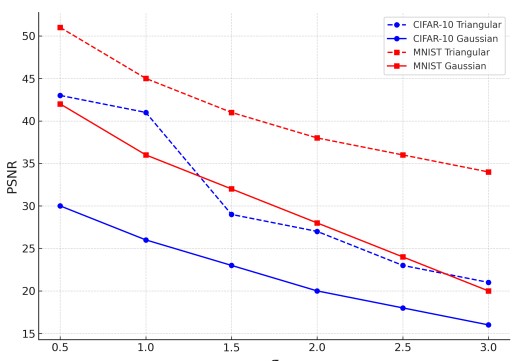

Figure 4: Ablation experiments of the lower bound with respect to the overlap coefficient, controlled by the noise standard deviation $\sigma$, on MNIST and CIFAR-10 with Gaussian and triangular noise.

### 5.1.3 BINARY LATENT AUTOENCODER

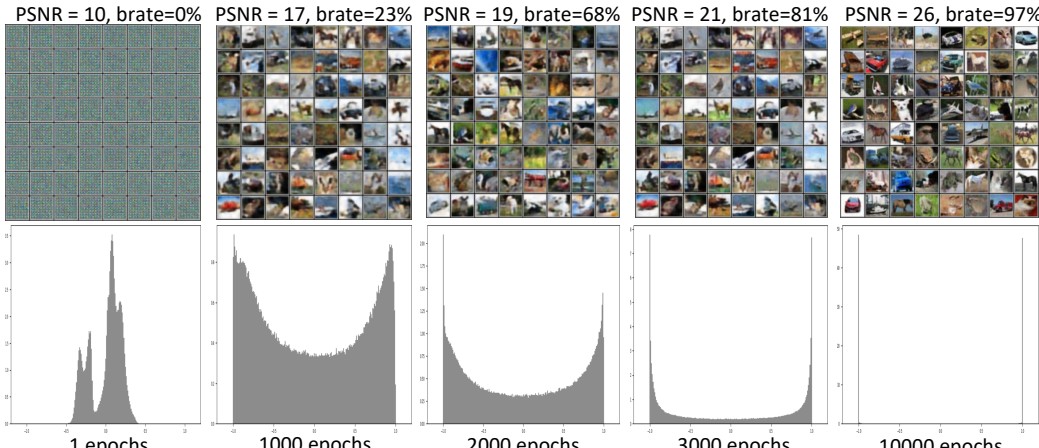

Figure 5: Demonstration of the binary rate of latent values (brate) and reconstruction quality under Peak Signal-to-Noise Ratio (PSNR) across training epochs. The histogram of latent values is illustrated on the second row.

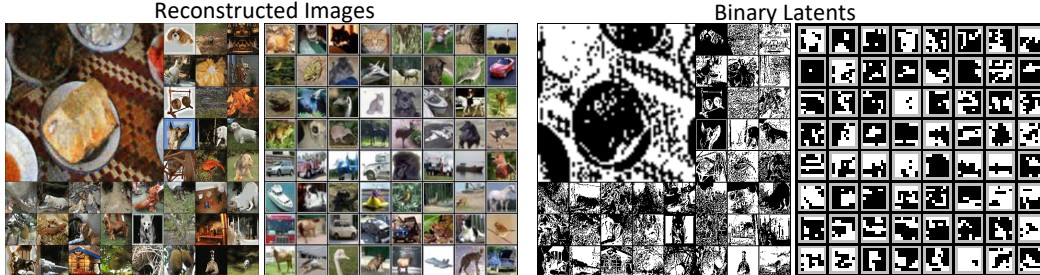

Figure 6: Qualitative evaluation of the proposed BL-AE on CIFAR-10 and 1k images subset of ImageNet.

Our Binary Latent Autoencoder (BL-AE) has a few advantages compared to existing State-of-the-Art Autoencoders for latent representation extraction, such as VQ-VAE (van den Oord et al., 2017), DC-AE (Chen et al., 2024), and SD-VAE (Rombach et al., 2022). First, BL-AE is able to capture discrete binary latent representations using a single reconstruction loss function (e.g., mean squared error). In contrast, VQ-VAE relies on an additional K-means clustering to learn a codebook. For demonstration, the values of latent variables converge to signed binary during training, as shown in Fig. 5. This is because our BL-AE is based on the Mutually Exclusive Probability Space. Second, the reconstruction quality of the proposed BL-AE correlates monotonically with the overlapping coefficient, which can be easily controlled by a parameter describing the intensity of noise, such as $\sigma$ (Fig. 4). Last but not least, as the latent values are binary, the latents provided by the proposed model require significantly less memory. As shown in Tab. 1, the total number of bits for DC-AE is $8 \times 8 \times 32 \times 32 = 65,536$ bits, whereas BL-AE only needs $16 \times 12 \times 1 = 192$ bits. To further illustrate the visualization results, we encode subsets of the CIFAR and ImageNet datasets using latent sizes of $16 \times 16 \times 1$ and $64 \times 64 \times 1$, respectively. The qualitative results of our binary latent representation are shown in Fig. 6.

Table 1: Comparison between the proposed Binary Latent Autoencoder with state-of-the-art Autoencoders including DC-AE (Chen et al., 2024) and SD-VAE (Rombach et al., 2022) in CIFAR-10 dataset.

| Method | Latent Shape | rFID $\downarrow$ | PSNR $\uparrow$ |
|---|---|---|---|
| DC-AE$_1$ | $8 \times 8 \times 32 \times 32$ bits | 1.08 | 26.41 |
| DC-AE$_2$ | $4 \times 4 \times 128 \times 64$ bits | 2.30 | 28.71 |
| SD-VAE$_1$ | $8 \times 8 \times 32 \times 32$ bits | 6.81 | 19.01 |
| SD-VAE$_2$ | $4 \times 4 \times 128 \times 64$ bits | 8.53 | 22.34 |
| Our BL-AE | $4 \times 4 \times 12 \times 1$ bits | 0.006 | 38.12 |

## 5.2 LOCAL DEPENDENCE HYPOTHESIS

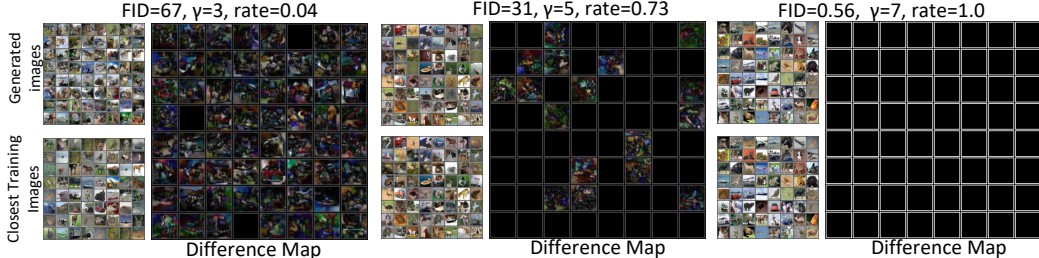

Figure 7: Images generated by our ARVM with observation range $\gamma=7$, 5, and 3.

The main focus of this paper is to investigate whether learning global distribution leads to memorization. We propose LDH as the mathematical framework for our $\gamma$-ARVM. The proposed $\gamma$-ARVM is able to learn distributions with a variable observation range $\gamma$. We, therefore, have the theoretical and experimental tool for our investigation. We first capture latents by our BL-AE with the architecture in Tab. 5, with the latents of size $N \times 8 \times 4 \times 4$. In particular, the CIFAR-10 dataset is utilized. Then by setting the observation ranges as 3, 5, and 7, we observe that ARVM

becomes a pure observation model, which achieves very good FID values (Sec. A.3). The results are shown in Fig. 7. For each observation range setting, we compute the memorization rate. In particular, for every generated image, we compute PSNR with its closest training image. When the PSNR value is greater than 30, we consider that they are the same image, following common practice in image quality assessment. Hence, we also utilize the PixelCNN (van den Oord et al., 2016) to reproduce our experiments, with a popular implementation on GitHub. The results are shown in Fig. 8. In particular, when we first directly input the latents into PixelCNN, we did not observe a strong memorization, which is the result of PixelCNN (v1). We then double the network's parameters and reduce the number of images to 1k, with the same architecture. PixelCNN (v2) also becomes a memorization model. Since both the ARVM and PixelCNN achieve the same conclusion, our concern that learning global distribution tends to lead to memorization rather than generative behavior is verified. We further evaluate memorization on high-resolution datasets, including 1k images from ImageNet and CelebA-HQ. Considering the computational cost, we directly increase the observation range to the global distribution. In this setting, memorization emerges prominently, with over 90% memorization rate and competitive FID scores (Tab. 3).

### 5.3 ON THE NATURE OF DISTRIBUTIONS

The main reason for memorization in autoregressive models stems from a deeper philosophical question about the nature of probability distributions: what is a distribution? Is it an objective reality, or merely a subjective belief? The frequentist perspective views probability distributions as objective realities, defined by the long-run frequencies with which events occur over time. Thus, subjective prior assumptions should be strictly controlled, with minimal human intervention. From this standpoint, memorization is not a flaw, but rather a faithful reflection of the empirical distribution observed from finite data, and arguably the best available ap-

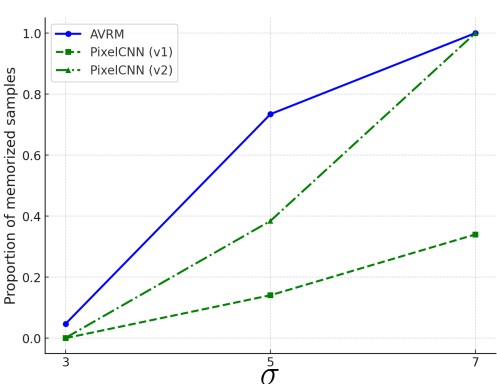

Figure 8: Demonstration of memorization rate with respect to the observation range $\gamma$ given values of 3, 5, 7.

proximation to the true distribution. Autoregressive models embody this frequentist perspective. However, in practical engineering, memorization is typically something to be avoided, and artificial prior assumptions are often introduced. This aligns with the Bayesian view, which treats probability distributions as subjective beliefs. Bayesian methods are therefore more flexible with prior assumptions, which is one of the main reasons why VAEs, GANs, and diffusion models employ a prior sampling distribution. For example, Gaussian priors in VAEs and diffusion models. Unfortunately, the reliance on prior assumptions introduces a high degree of subjectivity, and potentially even bias, into the evaluation and comparison of models. Under such circumstances, a clear gap emerges between scientific objectivity and engineering subjectivity. The proposed Local Dependence Hypothesis (LDH) can serve as a bridge to this gap. Since locality is assumed, autoregressive models are still able to perform generation rather than memorization.

## 6 CONCLUSION

In this work, we proposed two theoretical frameworks: 1) Mutually Exclusive Probability Space (MEPS) and 2) the Local Dependence Hypothesis (LDH). These frameworks were designed to investigate a potential limitation in probabilistic generative modeling; namely, learning global distributions tends to result in memorization rather than true generation. In particular, we focus on autoregressive models. MEPS motivated the development of the Binary Latent Autoencoder (BLAE), which encodes images into binary latent representations. These representations serve as input to our Autoregressive Random Variable Model (ARVM), which can be configured to model either global distributions or local dependences. When trained to model global distributions, ARVM becomes a memorization model. In contrast, when local dependences are emphasized, ARVM exhibits generative behavior, producing novel images by recombining learned features. Comprehensive experiments and discussions were conducted to support our hypotheses.

## 7 ETHICS STATEMENT

This work does not present any ethical concerns. The datasets used are publicly available and contain no sensitive information.

## 8 REPRODUCIBILITY STATEMENT

We have made efforts to ensure the reproducibility of our work. The network architectures utilized in this paper are provided in Tab. 4 and Tab. 5. raw data for normalization is provided in Tab. 6. Details of the experimental setup are described in Sec. A.3.1. Proofs of the mathematical derivations are presented in Sec. A.2. Source code and running scripts will be released upon acceptance of this paper.

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

# A APPENDIX

## A.1 DISCUSSION

### A.1.1 IMPROVING VAE WITH SMALLER STANDARD DEVIATION IN GAUSSIAN PRIOR

Based on the proposed Theorem 3.3, the reconstruction quality is closely related to the lower bound with respect to the overlap coefficient. An easy way to reduce the overlap coefficient is to use a smaller standard deviation $\sigma$; since with decreasing $\sigma$, the latent spaces will have more room to tolerate the overlap coefficient. To demonstrate this, we utilize a standard VAE implementation from GitHub, and gradually reduced the $\sigma$ of noise. Note that all other parts are kept invariant, including random seeds, optimization methods, and training epochs, etc. In particular, the MNIST dataset is utilized for evaluation. As shown in Fig. 9, the generation quality of the VAE increases with the decrease of $\sigma$.

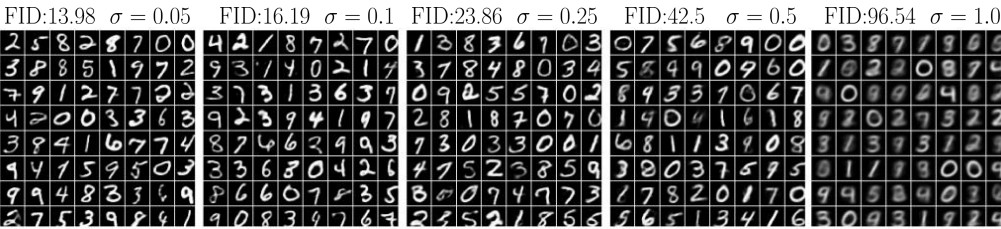

Figure 9: The generation quality of Variational Autoencoder can be improved by using a smaller standard deviation $\sigma$ in the Gaussian prior.

### A.1.2 MEMORIZATION IN VARIATIONAL AUTOENCODER MODELS

The fundamental assumption of VAEs is to encode image distribution into a latent distribution, with the ELBO used for optimization. The overlap coefficient in VAEs varies depending on the balance between the KL loss and the reconstruction loss. To adjust the overlap coefficient, we replace Gaussian noise with triangular noise and constrain the latent variables using the tanh function, ensuring that values in the latent space remain within the range $[-1, 1]$. Then by setting the $\sigma = 1$, we create a condition that each dimension of latent space is able to include 2 different latent random variables without creating MEPS. More specifically, two latent random variables centered at -1 and 1, with sigma as 1, so there are no overlap between the distributions of these two latents random variables. With the increase in the number of dimensions, the total possible number of random variables without OC becomes $2^M$, where M is the number of dimensions. As the number of dimensions increases, we obtain the results shown in Fig. 10. In the low-dimensional case, since the total space for the overlap coefficient is insufficient, the reconstruction is not similar to the training images. However, with an increasing number of dimensions, the reconstructed images gradually become closer to the training images.

### A.1.3 MEMORIZATION IN GENERATIVE ADVERSARIAL NETWORK

The overlap coefficient in Generative Adversarial Network is 1, since the input of generator is pure noise. In this condition, the GAN can be considered a mapping from latent variables shared the same expectation which is usually 0. Our idea to reduce overlap coefficient in GANs is to extend the input noise from a single distribution to a mixture distribution, such as a Gaussian mixture. Moreover, the means of Gaussian components are also parameterized to optimizable. During training, both the parameters of Gaussians, generator and discriminator are updated. We utilized 1k images from CIFAR-10 for experiments, with 20000 epochs used for training. In particular, the standard implementation in PyTorch of GAN is utilized. Expected extending the input from pure noise into optimizable Gaussians, all the remaining parts are kept invariant. The resulting figure is shown in Fig. 11. The loss of generator and discriminator are quite common compared to regular GANs, but every generated images are very similar to training images.

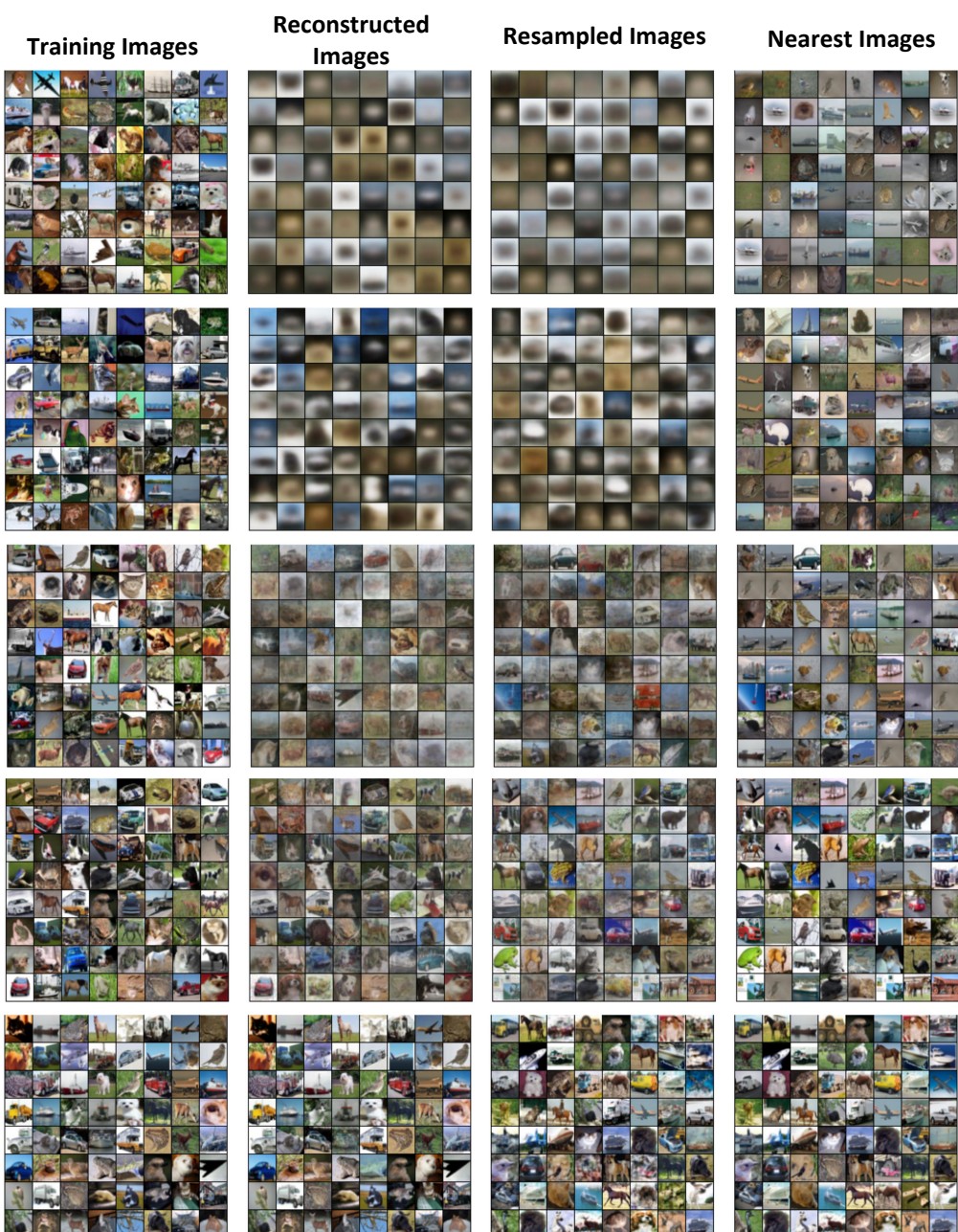

Figure 10: By using a triangular distribution and limiting the values of latent samples in range $[-1, 1]$ it is easy to create continuous probabilistic fields in high dimensional space. All samples in this fields will generate images that are very similar to training images.

### A.1.4 MEMORIZATION IN DIFFUSION MODELS

The easiest way to control the level of overlap coefficient is by reducing the number of training images. Therefore, we trained the DDPM Ho et al. (2020a) model on the CIFAR-10 dataset with varying numbers of training images: 16, 256, 1024, 2048, and 5120. To limit the fitting ability, we adopt a U-Net with only 9.27M learnable parameters as the backbone network for DDPM. The experiments are shown in Fig. 12. Specifically, the first row displays the generated results from trained models with varying numbers of training images: 16, 256, 1024, 2048, and 5120. The

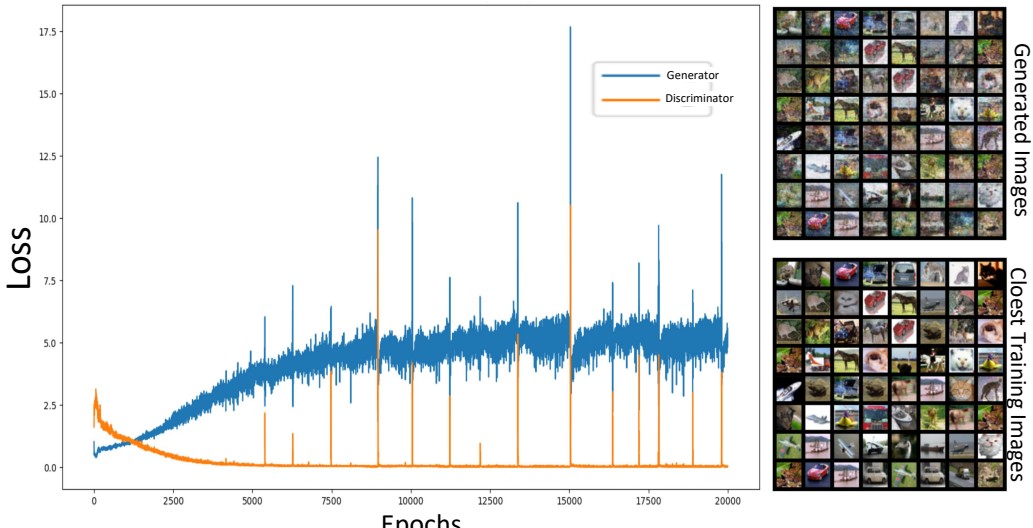

Figure 11: By extending the input of generator from pure noise to optimizable mixture of Gaussians noises, the GANs also tend to degrade into a memorization model.

second row shows the corresponding similar images in the training set. We employ the Structural Similarity Index (SSIM) to find the most similar images. As shown in Fig. 12, when the number of training images is 16, the diffusion model always outputs training images. When the number of training images increases to 1024, we can only see a few differences in the details. When the training images further increase to 5120, the diffusion model demonstrates images that are close to image fusion.

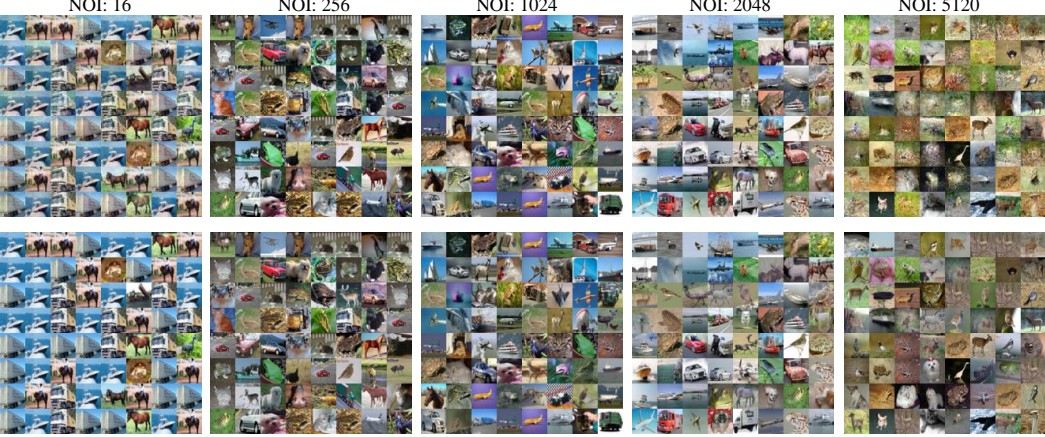

Figure 12: The DDPM model trained by different Number of Images (NOI). The images in the first row are generated images, while the images in the second row are the closest original images determined by Structural Similarity Index. We can observe that as the size of the training dataset increases, the generated images become less and less similar to the original images.

### A.1.5 OVERFITTING DISCUSSION

The original purpose of overfitting is to describe the gap between training and testing performance, which reflects generalization ability. In generation tasks, however, there is no single gold-standard target for a test set, although generalization can still be assessed on held-out data via proxy metrics (e.g., likelihood, FID, human evaluation). Consequently, equating overfitting with mere memorization is intuitively appealing but not strictly correct. Moreover, considering that the target of an

autoregressive model is to fit the empirical distribution, a vanishing training loss on finite data often increases the risk of memorization. Thus, some methods deliberately avoid over-optimization of the training objective as a form of regularization aimed at improving generalization performance. Unfortunately, from a mathematical-logical perspective, given that a loss function is designed to measure the discrepancy between a model and its target, the natural interpretation is that the global optimum corresponds to the loss approaching zero. In such a case, achieving zero loss should indicate that the model has perfectly captured the target distribution. However, in practice, the notion of overfitting is often introduced to suggest that a vanishing training loss reflects memorization rather than generalization. This raises a conceptual tension: if zero is not regarded as the true optimum, then how should one define the boundary between acceptable convergence and overfitting? Is approaching zero asymptotically still problematic, or only reaching it exactly? From this perspective, overfitting appears less as a logically necessary concept.

### A.1.6 FID COMPARISON FAIRNESS

A common concern could be the fairness of comparing the proposed ARVM with state-of-the-art generative models. Practically, it is true that such a comparison is "unfair," since the FID results of the proposed approach essentially come from memorization, while there is clear evidence that SOTA methods like diffusion are capable of generating novel images. However, from a mathematical-logical perspective, since no prior images are explicitly involved in the sampling steps, the FID of the proposed ARVM is still comparable. Logically speaking, even a purely memorization-based model still fits the minimal definition of a generative model. Of course, our goal is not to argue over semantics. The true issue lies in the evaluation metric: FID is insensitive to memorization. Moreover, since the proposed approach is basically an autoregressive model, there is no evidence to disprove that the FID reported by SOTA methods is not also benefiting from memorization, especially in autoregressive settings. Indeed, numerous recent works point to memorization in various generative paradigms, including diffusion (Carlini et al., 2023) and autoregressive models (Kowalczuk et al., 2025; Kasliwal et al., 2025; Yu et al., 2025). Ultimately, the main focus of the proposed approach is not to "beat" SOTA methods, but to encourage critical reflection on the core assumptions underlying generative modeling.

### A.2 MATHEMATICAL DERIVATION

### A.2.1 PROOF OF RECONSTRUCTION MSE LOWER BOUND IN THEOREM 3.3

Given a set of images $\mathbf{X} = \{\mathbf{x}_i\}_{i=1}^N$, encoder $e_\theta$ and decoder $d_\phi$. Let $\mathbf{z}_i = e_\theta(\mathbf{x}_i)$ and inject symmetric, unimodal noise to obtain $\tilde{\mathbf{z}}_i = \mathbf{z}_i + \epsilon_i$ with density $p_{\tilde{\mathbf{z}}_i}(\cdot)$. For any pair $(i,j)$ define:

$$p_m^{(i,j)}(\mathbf{z}) = \min\big(p_{\tilde{\mathbf{z}}_i}(\mathbf{z}),\, p_{\tilde{\mathbf{z}}_j}(\mathbf{z})\big). \tag{11}$$

Then:

$$\frac{1}{N}\sum_{i=1}^N \mathbb{E}\big[\|d(\tilde{\mathbf{z}}_i) - \mathbf{x}_i\|^2\big] = \frac{1}{2N^2}\sum_{i=1}^N\sum_{j=1}^N \Big(\mathbb{E}\|d(\tilde{\mathbf{z}}_i) - \mathbf{x}_i\|^2 + \mathbb{E}\|d(\tilde{\mathbf{z}}_j) - \mathbf{x}_j\|^2\Big)$$

$$= \frac{1}{2N^2}\sum_{i,j} \Big(\int p_{\tilde{\mathbf{z}}_i}(\mathbf{z})\|d(\mathbf{z}) - \mathbf{x}_i\|^2\, d\mathbf{z} + \int p_{\tilde{\mathbf{z}}_j}(\mathbf{z})\|d(\mathbf{z}) - \mathbf{x}_j\|^2\, d\mathbf{z}\Big)$$

$$= \frac{1}{2N^2}\sum_{i,j}\int p_m^{(i,j)}(\mathbf{z})\Big(\|d(\mathbf{z}) - \mathbf{x}_i\|^2 + \|d(\mathbf{z}) - \mathbf{x}_j\|^2\Big)\, d\mathbf{z}$$

$$+ \frac{1}{2N^2}\sum_{i,j}\zeta_{ij}(\mathbf{Z}, \mathbf{X}), \tag{12}$$

where the expression of $\zeta_{ij}(\mathbf{Z}, \mathbf{X})$ is:

$$\zeta_{ij}(\mathbf{Z}, \mathbf{X}) := \int \big(p_{\tilde{\mathbf{z}}_i}(\mathbf{z}) - p_m^{(i,j)}(\mathbf{z})\big)\|d(\mathbf{z}) - \mathbf{x}_i\|^2\, d\mathbf{z} + \int \big(p_{\tilde{\mathbf{z}}_j}(\mathbf{z}) - p_m^{(i,j)}(\mathbf{z})\big)\|d(\mathbf{z}) - \mathbf{x}_j\|^2\, d\mathbf{z}. \tag{13}$$

Let $A_{ij} = \{\mathbf{z} : p_{\tilde{\mathbf{z}}_i}(\mathbf{z}) \geq p_{\tilde{\mathbf{z}}_j}(\mathbf{z})\}$ and $B_{ij} = A_{ij}^\complement$. On $A_{ij}$, $p_{\tilde{\mathbf{z}}_i} - p_m^{(i,j)} = p_{\tilde{\mathbf{z}}_i} - p_{\tilde{\mathbf{z}}_j} \geq 0$ and $p_{\tilde{\mathbf{z}}_j} - p_m^{(i,j)} = 0$; on $B_{ij}$, the roles swap. Since the weights are nonnegative and the squared terms

are nonnegative, we have:

$$\boxed{\zeta_{ij}(\mathbf{Z}, \mathbf{X}) \geq 0}.$$

By the parallelogram inequality,

$$\|d(\mathbf{z}) - \mathbf{x}_i\|^2 + \|d(\mathbf{z}) - \mathbf{x}_j\|^2 \ \geq \ \tfrac{1}{2}\|\mathbf{x}_i - \mathbf{x}_j\|^2, \tag{14}$$

hence:

$$\frac{1}{N} \sum_{i=1}^{N} \mathbb{E}\big[\|d(\tilde{\mathbf{z}}_i) - \mathbf{x}_i\|^2\big] \ \geq \ \frac{1}{4N^2} \sum_{i,j} \|\mathbf{x}_i - \mathbf{x}_j\|^2 \int p_m^{(i,j)}(\mathbf{z})\, d\mathbf{z} \ + \ \frac{1}{2N^2} \sum_{i,j} \zeta_{ij}(\mathbf{Z}, \mathbf{X}). \tag{15}$$

Since $\int p_m^{(i,j)}(\mathbf{z})\, d\mathbf{z} = \mathrm{OC}(\tilde{\mathbf{z}}_i, \tilde{\mathbf{z}}_j)$ and $\zeta_{ij} \geq 0$, we obtain the lower bound

$$\boxed{\frac{1}{N} \sum_{i=1}^{N} \mathbb{E}\big[\|d(\tilde{\mathbf{z}}_i) - \mathbf{x}_i\|^2\big] \ \geq \ \frac{1}{4N^2} \sum_{i,j=1}^{N} \mathrm{OC}(\tilde{\mathbf{z}}_i, \tilde{\mathbf{z}}_j)\, \|\mathbf{x}_i - \mathbf{x}_j\|^2}.$$

**Remark A.1.** Although the above lower bound is derived under the squared error loss, the key structure does not rely on the specific quadratic form. The overlap coefficient $\mathrm{OC}(\tilde{\mathbf{z}}_i, \tilde{\mathbf{z}}_j)$ arises solely from the probabilistic overlap of the perturbed latent codes and is independent of the loss. The constant factor $\tfrac{1}{2}\|\mathbf{x}_i - \mathbf{x}_j\|^2$ in the bound originates from the parallelogram inequality in Eq. 14, which is a consequence of the strong convexity of the squared norm. For a general convex (or strongly convex) loss, one may obtain an analogous lower bound where the constant changes according to the convexity parameter of the chosen loss. Thus, the phenomenon that reconstruction error is fundamentally limited by the overlap coefficient in MEPS is not specific to the squared loss, but extends to a broader family of convex losses.

### A.2.2 Proof of Mutual Exclusivity in Theorem 3.5

The expression of Theorem 3.5 is shown as:

$$\underset{\mathbf{z}_i, \mathbf{z}_j}{\arg\min}\ \mathbb{E}_\epsilon[\mathrm{OC}(\tilde{\mathbf{z}}_i, \tilde{\mathbf{z}}_j)] \Rightarrow \underset{\mathbf{z}_i, \mathbf{z}_j}{\arg\min}\ \mathbb{E}_\epsilon[\mathrm{OC}(\mathbf{z}_i + \epsilon, \mathbf{z}_j + \epsilon)] \Rightarrow \underset{\mathbf{z}_i, \mathbf{z}_j}{\arg\max}\ \frac{1}{N^2} \sum_{i,j} \|\mathbf{z}_i - \mathbf{z}_j\|^2. \tag{16}$$

where noise $\epsilon$ is a symmetric, unimodal function with $f(\cdot)$ as its probability density function. Therefore, the probability density function of $\tilde{\mathbf{z}}_i$ and $\tilde{\mathbf{z}}_j$ is:

$$p_{\tilde{\mathbf{z}}_i}(\mathbf{z}) = f(\mathbf{z} - \mathbf{z}_i), \quad p_{\tilde{\mathbf{z}}_j}(\mathbf{z}) = f(\mathbf{z} - \mathbf{z}_j). \tag{17}$$

Then, by plugging this expression into the overlap coefficient in Eq. 1, we have:

$$\mathrm{OC}(\tilde{\mathbf{z}}_i, \tilde{\mathbf{z}}_j) = \int \min\big(p_{\tilde{\mathbf{z}}_i}(\mathbf{z}),\, p_{\tilde{\mathbf{z}}_j}(\mathbf{z})\big)\, d\mathbf{z} = \int \min\big(f(\mathbf{z} - \mathbf{z}_i),\, f(\mathbf{z} - \mathbf{z}_j)\big)\, d\mathbf{z}. \tag{18}$$

Since $f$ is symmetric and radially unimodal (e.g., Gaussian), the overlap coefficient $\mathrm{OC}(\tilde{\mathbf{z}}_i, \tilde{\mathbf{z}}_j)$ depends solely on the Euclidean distance $d_{ij} = \|\mathbf{z}_i - \mathbf{z}_j\|$. Then we have:

$$\mathrm{OC}(\tilde{\mathbf{z}}_i, \tilde{\mathbf{z}}_j) = h(\|\mathbf{z}_i - \mathbf{z}_j\|), \tag{19}$$

where $h(\cdot)$ is a strictly decreasing function. Therefore, minimizing the sum of all pairwise overlaps is equivalent to minimizing the sum over all $h(d_{ij})$. Since $h(\cdot)$ is strictly decreasing, this objective is effectively enforced by maximizing the pairwise distances $d_{ij} = \|\mathbf{z}_i - \mathbf{z}_j\|$. Theorem 3.5 is thus proved.

**Remark A.2.** This proof shows that minimizing overlap between symmetric, unimodal latent distributions is mathematically equivalent to maximizing their pairwise distances. The main limitation of this proof is the reliance on symmetric, unimodal assumptions, which may not extend to more complex or multimodal priors. However, as the noise is usually injected into the elements of tensors, such a proof is sufficient for analyzing our MEPS in variable generative models like VAEs, GANs, and diffusion.

### A.2.3 On the Monotonicity of OC with Respect to Scale

For the two distributions we use (Gaussian and symmetric triangular), the overlap coefficient (OC) between two shifted copies with fixed mean separation $\Delta$ increases monotonically with the scale parameter $\sigma$.

**Gaussian case:** For $X \sim \mathcal{N}(\mu_1, \sigma^2)$ and $Y \sim \mathcal{N}(\mu_2, \sigma^2)$ with $\Delta = |\mu_1 - \mu_2|$, the overlap coefficient is

$$\mathrm{OC}(\sigma, \Delta) = 2\,\Phi\!\left(-\tfrac{\Delta}{2\sigma}\right), \tag{20}$$

where $\Phi$ is the standard Gaussian CDF. Differentiating gives

$$\frac{\partial \mathrm{OC}}{\partial \sigma} = 2\,\phi\!\left(\tfrac{\Delta}{2\sigma}\right) \cdot \tfrac{\Delta}{2\sigma^2} \;>\; 0, \tag{21}$$

so OC increases strictly with $\sigma$.

**Triangular case:** For triangular distribution, we set $\kappa = 2$, centered at $\mu_1, \mu_2$ with half-width $\sigma$, the overlap region is the intersection of two isosceles triangles. Its area is a quadratic function of the overlap length, which grows linearly with $\sigma$. A direct calculation gives

$$\mathrm{OC}(\sigma, \Delta) \;=\; \begin{cases} \left(1 - \tfrac{\Delta}{2\sigma}\right)^2, & 0 \le \Delta \le 2\sigma, \\ 0, & \Delta \ge 2\sigma. \end{cases} \tag{22}$$

Clearly, $\frac{\partial \mathrm{OC}}{\partial \sigma} > 0$ whenever overlap exists.

**Remark A.3.** Thus, in both Gaussian and triangular settings, scaling $\sigma$ monotonically enlarges the overlap for fixed $\Delta$, justifying our use of $\sigma$ as a practical proxy for controlling OC.

### A.2.4 OC=1 in GANs

For GANs, the generator input is pure noise. During training, the number of input noise vectors usually equals the batch size, and each noise vector is mapped through the generator to produce a fake image. In our MEPS framework, these input noise vectors can be regarded as a set of random variables. Specifically, drawing n samples from the same distribution is equivalent to defining n random variables that follow the same distribution with identical expectation and sampling each once. Under this view, all input variables in GANs share the same distribution and expectation, and thus their overlap coefficient (OC) equals 1. This situation corresponds to an extreme case in MEPS where all random variables completely overlap. Intuitively, this means that the model lacks any separation margin during training, making the optimization more unstable. We believe this perspective offers an explanation for the well-known training difficulties of GANs. It should be emphasized that this is not a formal proof, but rather an interpretative understanding.

### A.3 Diagnostic Evaluation: Comparison with State-of-the-Art Methods

We also compare our $\gamma$-Autoregressive Random Variable Model (ARVM), with observation range of 7, 5, and 3, to get the FID scores of 7-ARVM, 5-ARVM and 3-ARVM shown in Tab. 2 and Tab. 3. In particular, the architecture is described in Tab. 5, with NoD=32. Since the spatial size of binary latent is $4 \times 4$, our ARVM learns the global distribution when observation range = 7 (padding is used when spatial size is too small.) 7-ARVM achieves an FID score of 0.56 when learning global distributions, which often results in training-sample memorization. Unfortunately, such FID scores remain comparable to those of state-of-the-art methods under identical evaluation conditions (Sec. A.1.6). Notably, this is achieved without relying on any prior assumptions related to image structure during the sampling process. Similar results are observed on high-resolution datasets including ImageNet, CelebA-HQ, and LSUN Bedroom, as shown in Tab. 3. The main reason for such low FID is primarily the memorization effect in the proposed $\gamma$-ARVM. However, since the proposed ARVM is essentially a standard autoregressive model, especially when the observation range is increased to learn global distributions, it is worth considering that the current claims that autoregressive models outperform diffusion models may simply be a consequence of memorization (Sun et al., 2024; Zhang et al., 2025). Likewise, it is also worth considering that the reported superiority of diffusion over VAEs or GANs may be due to the same reason.

Table 2: Diagnostic Evaluation on CIFAR-10 Using FID and Inception Scores on the CIFAR-10 dataset.

| | Method | FID score ↓ | Inception Score↑ |
|---|---|---|---|
| **Diffusion** | DDPM (Ho et al., 2020b) | 3.17 | $9.46 \pm 0.11$ |
| | EDM (Cui et al., 2023) | 1.30 | N/A |
| **GAN** | CCF-GAN (Li et al., 2023) | 6.08 | N/A |
| | KD-DLGAN (Cui et al., 2023) | 8.30 | N/A |
| | StyleGAN2 (Karras et al., 2020) | 3.26 | $9.74 \pm 0.05$ |
| | SN-SMMDGAN (Arbel et al., 2018) | 25.00 | 7.30 |
| **VAE** | NCP-VAE (Aneja et al., 2021) | 24.08 | N/A |
| | NVAE (Vahdat & Kautz, 2020) | 32.53 | N/A |
| | DC-VAE (Parmar et al., 2021) | 17.90 | 8.20 |
| | NCSN (Song & Ermon, 2019) | 25.32 | 8.87 |
| **Ours** | $ARVM_3$ | 67.13 | $6.32 \pm 0.23$ |
| | $ARVM_2$ | 31.42 | $7.12 \pm 0.15$ |
| | $ARVM_1$ | 0.56 | $11.15 \pm 0.13$ |

Table 3: Diagnostic Evaluation on CIFAR-10 Using FID and Inception Scores on high-resolution datasets.

| Dataset | Model | Method | FID ↓ |
|---|---|---|---|
| **LSUN Bedroom** | **Diffusion** | DDPM (Ho et al., 2020b) | 6.36 |
| | **GAN** | PGGAN (Karras et al., 2018) | 8.34 |
| | | PG-SWGAN (Wu et al., 2019) | 8.00 |
| | **Ours** | $ARVM_1$ | 1.54 |
| **ImageNet** | **Diffusion** | DiT-XL/2 (Peebles & Xie, 2023) | 9.62 |
| | | DiT-XL/2-G (Peebles & Xie, 2023) | 2.27 |
| | **Transformer** | MaskGIT (Chang et al., 2022) | 6.18 |
| | | VQGAN+Transformer (et al., 2021) | 6.59 |
| | **Ours** | $ARVM_1$ | 5.63 |
| **CelebA-HQ 256x256** | **VAE** | NVAE (Vahdat & Kautz, 2020) | 48.27 |
| | **Ours** | $ARVM_1$ | 1.53 |

### A.3.1 EXPERIMENTAL DETAILS

All experiments were conducted on a single RTX 4090 GPU with 24 GB of VRAM. Training and testing for each experiment were completed within 24 hours on this single GPU, given the computational constraints. For the same reason, large-scale experiments on larger models were not feasible. All implementations were based on PyTorch, and the Adam optimizer was used for training. Source code and running scripts will be released upon acceptance of this paper.

### A.4 THE USE OF LARGE LANGUAGE MODELS (LLMS)

We utilized Grammarly and ChatGPT solely to check typos and grammar in the proposed paper. No technical content, experiments, or analysis were generated by large language models.

Table 4: Details of our network architecture

| | Type | weight | stride | padding | Data size |
|---|---|---|---|---|---|
| **Encoder** | Input | | | | $N \times 3 \times 32 \times 32$ |
| | Conv2d | $64 \times 3 \times 4 \times 4$ | 2 | 1 | $N \times 64 \times 16 \times 16$ |
| | LeakyReLU | | | | |
| | Conv2d | $256 \times 64 \times 4 \times 4$ | 2 | 1 | $N \times 256 \times 8 \times 8$ |
| | LeakyReLU | | | | |
| | Conv2d | $256 \times 1024 \times 1 \times 1$ | 1 | 0 | $N \times 1024 \times 8 \times 8$ |
| | Conv2d | $1024 \times \text{NoD} \times 1 \times 1$ | 1 | 0 | $N \times \text{NoD} \times 8 \times 8$ |
| **Latents** | | | | | $N \times \text{NoD} \times 8 \times 8$ |
| **Decoder** | Linear | $\text{NoD} \times 1024 \times 1 \times 1$ | 1 | 0 | $N \times 1024 \times 8 \times 8$ |
| | Linear | $1024 \times 1024 \times 1 \times 1$ | 1 | 0 | $N \times 1024 \times 8 \times 8$ |
| | LeakyReLU | | | | |
| | ConvT2d | $512 \times 1024 \times 3 \times 3$ | 3 | 1 | $N \times 512 \times 8 \times 8$ |
| | LeakyReLU | | | | |
| | ConvT2d | $64 \times 512 \times 4 \times 4$ | 2 | 1 | $N \times 64 \times 16 \times 16$ |
| | ConvT2d | $3 \times 64 \times 4 \times 4$ | 2 | 1 | $N \times 3 \times 32 \times 32$ |
| | Tanh | | | | |
| **Refine** | Conv2d | $32 \times 3 \times 1 \times 1$ | 3 | 1 | $N \times 32 \times 32 \times 32$ |
| | LeakyReLU | $\alpha = 0.01$ | | | $N \times 32 \times 32 \times 32$ |
| | Conv2d | $3 \times 32 \times 1 \times 1$ | 3 | 1 | $N \times 3 \times 32 \times 32$ |
| | Output | | | | $N \times 3 \times 32 \times 32$ |

NoD: number of dimension.

Table 5: Details of our network architecture.

| | Type | weight | stride | padding | Data size |
|---|---|---|---|---|---|
| **Encoder** | Input | | | | $N \times 3 \times 32 \times 32$ |
| | Conv2d | $64 \times 3 \times 4 \times 4$ | 2 | 1 | $N \times 64 \times 16 \times 16$ |
| | LeakyReLU | | | | |
| | Conv2d | $256 \times 64 \times 4 \times 4$ | 2 | 1 | $N \times 256 \times 8 \times 8$ |
| | LeakyReLU | | | | |
| | Conv2d | $512 \times 256 \times 4 \times 4$ | 2 | 1 | $N \times 512 \times 4 \times 4$ |
| | LakyReLU | | | | |
| | Conv2d | $512 \times 8196 \times 1 \times 1$ | 1 | 0 | $N \times 8196 \times 4 \times 4$ |
| | Conv2d | $8196 \times \text{NoD} \times 1 \times 1$ | 1 | 0 | $N \times \text{NoD} \times 4 \times 4$ |
| **Latents** | | | | | $N \times \text{NoD} \times 4 \times 4$ |
| **Decoder** | Linear | $\text{NoD} \times 8196 \times 1 \times 1$ | 1 | 0 | $N \times 8196 \times 4 \times 4$ |
| | Linear | $8196 \times 1024 \times 1 \times 1$ | 1 | 0 | $N \times 1024 \times 4 \times 4$ |
| | LeakyReLU | | | | |
| | ConvT2d | $512 \times 1024 \times 4 \times 4$ | 1 | 0 | $N \times 512 \times 4 \times 4$ |
| | LeakyReLU | | | | |
| | ConvT2d | $256 \times 512 \times 4 \times 4$ | 2 | 1 | $N \times 256 \times 8 \times 8$ |
| | ConvT2d | $64 \times 256 \times 4 \times 4$ | 2 | 1 | $N \times 64 \times 16 \times 16$ |
| | ConvT2d | $3 \times 64 \times 4 \times 4$ | 2 | 1 | $N \times 3 \times 32 \times 32$ |
| | Tanh | | | | |
| **Refine** | Conv2d | $32 \times 3 \times 1 \times 1$ | 3 | 1 | $N \times 32 \times 32 \times 32$ |
| | LeakyReLU | $\alpha = 0.01$ | | | $N \times 32 \times 32 \times 32$ |
| | Conv2d | $3 \times 32 \times 1 \times 1$ | 3 | 1 | $N \times 3 \times 32 \times 32$ |
| | Output | | | | $N \times 3 \times 32 \times 32$ |

NoD: number of dimension.

Table 6: Min-Max normalization parameters settting.

|  | CIFAR-10 Tri. | MNIST Tri. | CIFAR-10 Gau. | MNIST Gau. |
|---|---|---|---|---|
| min meanDist | 28.01 | 36.11 | 50.76 | 72.43 |
| max meanDist | 1018.79 | 1131.68 | 1302.75 | 1068.97 |
| min minDist | 25.78 | 25.26 | 23.72 | 20.11 |
| max minDist | 806.93 | 315.31 | 988.17 | 414.71 |
| min recMSE | 6.97 | 2.18 | 6.43 | 1.89 |
| max recMSE | 296.37 | 36.31 | 395.81 | 45.13 |

Tri. Triangular noise, Gau. Gaussian noise.

