# OpenReview forum: "Exploring Image Generation via Mutually Exclusive Probability Spaces and Local Dependence Hypothesis"
_ICLR.cc/2026/Conference — ICLR 2026 Conference Withdrawn Submission_

### Official Review · Reviewer_kk8R · 2025-10-30

**Soundness:** 1
**Presentation:** 1
**Contribution:** 1
**Rating:** 0
**Confidence:** 4

**Summary:**

This paper attempts to test a widely assumed premise in probabilistic generative modeling for image generation, namely that learning the global data distribution is sufficient for sampling novel images. The goal is to assess whether training leads to genuine generalization or merely memorization of the training set. To examine this question, the paper introduces two theoretical frameworks, Mutually Exclusive Probability Space (MEPS) and the Local Dependence Hypothesis (LDH). Building on insights from these frameworks, the authors propose the Binary Latent Autoencoder (BL-AE) and the $\gamma$-Autoregressive Random Variable Model ($\gamma$-ARVM), and they evaluate these models through numerical experiments.

I have tried to understand the contribution without relying on subjective or biased judgment as much as I can. Unfortunately, to the best of my understanding, I am unable to identify the contribution of this work. Specific reasons are listed below.

**Strengths:**

- The problem formulation that training of probabilistic generative models might converge to overfitting on a finite dataset rather than proper generalization is interesting and of broad community interest.
- The research direction to seek general theory that can apply across commonly used probabilistic generative models such as VAEs, GANs, and diffusion models is valuable.

**Weaknesses:**

Overall, the paper is very difficult to understand. There are many unclear points in the mathematical formalization, and some theoretical claims are likely incorrect. Even for statements that may be correct, key assumptions used in the proofs are not properly stated in the main text. The exposition lacks logical consistency, and the discussion proceeds without sharing sufficient prerequisites for the reader to follow. As a whole, the argument breaks down. In sum, the present form does not allow a fair evaluation of the paper’s contribution. Representative issues are as follows.

- **Unclear logical development of the narrative:** In both the abstract and the introduction, it is hard to discern what the authors want to claim. For example, the following sentence appears at line 51

    > While this phenomenon is often attributed to overfitting, we argue that it is related to the core assumption of learning global distributions. The issue lies in differing philosophical views between the frequentist and Bayesian interpretations of whether probability distributions objectively exist.

    This statement lacks coherence with the surrounding context, and it is unclear why the frequentist vs Bayesian interpretational issue suddenly appears here. Beyond this representative example, the entire text exhibited unclear logical progression, making it extremely difficult to discern what the paper was trying to state.

- **Insufficient discussion of related work:** The question of whether generative models generalize to the true distribution rather than overfitting to the training data is important. However, the paper inadequately reviews prior work on this topic. For example, the following representative theoretical analyses for diffusion models are known and should be situated properly.
    - Oko, K., Akiyama, S., and Suzuki, T. Diffusion Models are Minimax Optimal Distribution Estimators. ICML 2023
    - Kadkhodaie, Z., Guth, F., Simoncelli, E. P., and Mallat, S. Generalization in diffusion models arises from geometry-adaptive harmonic representations. ICLR 2024
- **Unclear problem setup:** In Section 3.1, $\tilde{\mathbf{z}}_i \in \tilde{Z}$ is defined as a random variable, but in the subsequent development the paper alternates between treating this quantity as a random variable and as a deterministic variable, which breaks the logical consistency.

    For instance, at line 150 of the same section, the paper introduces $d_{\phi}: \tilde{Z} \to X$ as a deterministic mapping. This presupposes that $\tilde{Z}$ is a space of deterministic variables. If $\tilde{Z}$ is a set of random variables, then the image under $d_{\phi}$ would also be random variables, yet the paper defines $X = \\{\mathbf{x}_i\\}$ as a set of deterministic variables. In Section 4.1, the meaning of $\tilde{\mathbf{z}}_i$ clearly changes across the two sides of an  $\implies$ symbol. On the left, $\tilde{\mathbf{z}}_i$ is treated as deterministic, while on the right the argument uses mutual information, which is not computable unless $\tilde{\mathbf{z}}_i$ is random.

    Next, there is no explanation of what concrete objects $\tilde{\mathbf{z}}$ and $\mathbf{x}$ are intended to model. It is also unclear why these variables are modeled as a finite set $\tilde{Z} \coloneqq \\{\tilde{\mathbf{z}}_i\\}$ rather than elements of a continuous space, and what justifies that choice. In particular, it is unclear why each point is associated with a density and what the support of the density $p\_{\mathbf{z}\_i}(\mathbf{z})$ is.

- **Vague definition of MEPS:** One of the core concepts, MEPS, is defined in Definition 3.1, but the given text does not read as a proper definition and leaves the nature of MEPS unclear. The sentence “$(\tilde{Z}, d_{\phi})$ forms a MEPS if …” admits two interpretations
    - Using $(\tilde{Z}, d_{\phi})$ one can introduce an MEPS.
    - $(\tilde{Z}, d_{\phi})$ IS the definition of MEPS itself, that is, $\mathrm{MEPS} \coloneqq (\tilde{Z}, d_{\phi})$.

    In the former case, the definition does not explain how the pair introduces a MEPS. In the latter case, naming this mathematical object a “space” or “probability space” departs from common usage for the following reasons:

    - In mathematics, a space typically consists of a set together with structures on that set. From the notation, $d_{\phi}$ seems to provide structure, but since $d_{\phi}: \tilde{Z} \to X$ is merely a map between finite sets, it does not obviously endow $\tilde{Z}$ with any structure.
    - If the term “probability space” is used, one expects a triple such as sample space, event space, and probability function, or equivalent information.
- **Concerns about the validity of Theorem 3.3:**
    - The proof in the appendix suddenly introduces implicit assumptions that are never stated. The main text suggests that the claim holds for general pairs $\tilde{Z}$ and $X$, yet the proof assumes that these variables are linked by an encoder-decoder structure, and the argument essentially relies on that assumption. Theorem 3.3 does not appear to hold as stated. All assumptions used in the proof should be explicitly presented.
    - The quantity discussed in the proof (left-hand side of Equation 3) is called the mean squared error (MSE) in the paper, and the text debates whether its lower bound can be strictly positive. This discussion is misleading. The MSE defined in the theorem includes an expectation with respect to noise injected in the latent space, so it is in general natural for its infimum to be strictly greater than zero.
- **Error in the derivation of Theorem 3.5:** Equation 4, presented as Theorem 3.5, does not hold in general. The equation asserts implications $P \implies Q$ among three statements. The proof for the implication between the second and the third statements is incorrect. The proof argues that because the overlap coefficient (OC) introduced in this paper can be expressed as a monotonically decreasing function $h(\|\mathbf{z}_i - \mathbf{z}_j\|)$ of the Euclidean distance between two variables, the optimization reduces to maximizing the average of squared Euclidean distances. But in general,

    $$
    \operatorname{argmin}\_{ \\{ d\_{ij} \\}\_{i,j=1}^N } \sum\_{i,j} h(d\_{ij}) \neq \operatorname{argmax}\_{ \\{ d\_{ij} \\}\_{i,j=1}^N } \sum\_{i,j} d\_{i,j}^2
    $$

    for an arbitrary monotonically decreasing function $h$.

- **Unclear relationship to VAEs, GANs, and diffusion models:** The paper claims that handling OC in a certain way allows a unifying treatment of these models, but the logic is opaque. At a minimum, the theorems in Section 3 appear, as noted above, to tacitly assume an encoder-decoder architecture, whereas GANs do not usually employ an encoder. This point requires proper discussion.
- **Concerns about the definition of the noise distribution:** The definition in Equation 6 is unclear. The range (or codomain) of $\operatorname{Tri}(\kappa)$, which presumably denotes a probability density, clearly extends to negative values, and the density computation itself contains a random variable.
- **Justification of LDH for autoregressive models:** The paper posits Local Dependence Hypothesis (LDH) as a relationship between distances in latent space and mutual information (Equation 7), but the discussion proceeds without clarifying whether the latent variables are deterministic or random, which obscures the intended claim. Even if one accepts the premise of Equation 7, it is unclear how the assumption about the generative process for an autoregressive model in Equation 8 follows. According to Equation 7, $\gamma$ should represent a lower bound on distances in latent space, yet Equation 8 performs subtraction between that constant and the index of the inference step. These quantities have incompatible units under dimensional analysis.

**Questions:**

- What concrete objects are $\tilde{\mathbf{z}}$ and $\mathbf{x}$ intended to model? From the text I infer that they correspond to latent variables and images, respectively. Why are these variables modeled as elements of finite sets rather than points in continuous spaces? On what observations is the possible mismatch in the cardinalities of these sets based?
- This paper defines a density $p_{\mathbf{z}_i}(\mathbf{z})$ associated with each $\tilde{\mathbf{z}}_i$. What is the support of these probability densities? A literal reading of the definition would suggest that the finite set $\tilde{Z}$ serves as the support, but the subsequent development in the paper appears inconsistent with such an interpretation.
- Please provide a rigorous definition of MEPS.
- Please formally state the problem setup on which Theorem 3.3 relies, including any implicit assumptions.
- What does the right-hand side of Equation 6 represent? Please clarify its intended meaning.

---

### Official Review · Reviewer_ipYY · 2025-11-01

**Soundness:** 3
**Presentation:** 2
**Contribution:** 3
**Rating:** 4
**Confidence:** 3

**Summary:**

The paper challenges the prevailing belief that learning a global data distribution is sufficient for high-quality generation. It introduces two theoretical frameworks: mutually exclusive probability spaces (MEPS) and local dependency hypothesis (LDH) to analyze this assumption. MEPS demonstrates that deterministic mappings coupled with stochastic variables reduce distributional overlap deriving a lower bound between reconstruction error and overlap, and motivates the binary latent autoencoder (BL-AE), which encodes images into binary latent tokens for autoregressive modeling. LDH constrains autoregressive dependencies within a local radius γ, leading to the γ-autoregressive visual model (γ-ARVM) that predicts a histogram of possible next tokens instead of a single label, capturing multimodality and uncertainty explicitly. Experiments show that as γ increases (approaching global dependency), the model shifts from compositional generation to memorization, validating the hypothesis that global distribution learning tends to induce memory-like behavior. Overall, the paper theoretically and empirically links distribution overlap, reconstruction fidelity, and memorization, providing a foundation for understanding the trade-off between local generation and global memorization in deep generative models.

**Strengths:**

* The paper reframes image generation as local-dependence modeling rather than global density estimation, introduces a computable overlap/exclusivity formalism with a reconstruction error lower bound, and combines the bl-ae with γ-ARVM to yield a continuous local to global experimental knob. So the framework can serve as a diagnostic framework to distinguish genuine modeling from memorization.

**Weaknesses:**

* ARVM prediction head is under-specified. The paper defines only inputs/outputs and the histogram loss. It omits the backbone: layer types, masking or windowing, neighborhood distance, sequence linearization, depth/width, and parameter count. Training hyperparameters are also missing. Detailed tables are given only for the BL-AE encoder/decoder.
* “Token” definition and ordering are unclear. The paper does not explain how the 2-D latent grid is linearized. It does not state how borders or padding are handled. It does not define the distance metric for the γ neighborhood.
* Training protocol for γ is unclear. The paper does not say whether a separate model is trained for each γ. It also does not say whether they train once at a large γ and mask at inference. Without this, γ-sweep results are hard to interpret.
* ImageNet comparisons are somewhat misleading. Table 3 lists 256×256 pixel-space baselines. The paper does not state the resolution of its own ImageNet runs (pixel or latent), the subset size, or the FID sample count. The numbers are shown side-by-side as if they were matched. With a single RTX 4090 and a 1 day budget, a full ImageNet-256 training is implausible, so the reported FID is hard to interpret.
* The overall writing is very hard to follow. the exposition is dense with key definitions scattered and notation heavy.

**Questions:**

* How do MEPS and LDH relate? Which claims are theoretical (e.g., bounds), and which are empirical (e.g., γ-sweep effects)?
* Why is BL-AE necessary for γ-ARVM? What fails if we use other discrete tokens (e.g., small-K VQ or VAE with binning)?
* What exactly is a token here (bit-rate, spatial resolution, ordering), and how does γ act on this tokenization?
* For ImageNet: what resolution is used (pixel vs. latent), what subset size, and how many samples feed FID?
* Are comparisons matched to 256×256 pixel-space baselines, or are they diagnostic numbers on different settings?
* What is the training protocol vs. γ? Do you retrain per γ, or train once at large γ and mask at inference?
* What is the AR head architecture and size (layers, masking/windowing, params), and what are the key hyperparameters?
* What are the efficiency numbers (tokens/sec, wall-clock per sample) at the reported configurations?
* How is memorization audited beyond a nearest-train PSNR threshold (e.g., train/test NN in LPIPS/CLIP, membership inference)?

---

### Official Review · Reviewer_uyaq · 2025-11-01

**Soundness:** 2
**Presentation:** 2
**Contribution:** 2
**Rating:** 2
**Confidence:** 4

**Summary:**

This paper investigates a potential limitation in probabilistic generative models, arguing that the common goal of learning a global data distribution can lead to memorization rather than true generative behavior. The authors introduce two theoretical frameworks: the Mutually Exclusive Probability Space (MEPS), which posits that learning dynamics in deterministic mappings tend to reduce the overlap between latent variable distributions, and the Local Dependence Hypothesis (LDH), which suggests that generative capacity stems from modeling local rather than global dependencies. Based on these frameworks, the paper proposes Binary Latent Autoencoder (BL-AE), which leverages MEPS to learn compressed binary latents, and $\gamma$-Autoregressive Random Variable Model ($\gamma$-ARVM), which has a variable observation range $\gamma$ to test the LDH.

**Strengths:**

The proposed BL-AE achieves high reconstruction quality with a highly compressed latent space, demonstrating an alternative method for learning discrete representations without complex mechanisms like VQ.

**Weaknesses:**

1. The paper's central claim that "learning global distributions results in memorization rather than generative behavior" is presented as a general principle but is only supported by a narrow set of experiments on a specific type of autoregressive model (i.e. PixelCNN). The $\gamma$-LDA is proposed as the key to avoiding memorization, but it is just one of many possible approaches (e.g., regularization, inductive biases).
2. The paper frequently extends its analysis and conclusions, developed from an autoencoder and autoregressive perspective, to other generative model families like GANs and diffusion models without adequate justification. For instance, the claim that "degraded reconstruction fidelity leads to lower generation quality" (Lines 197-199) is a core assumption derived from the MEPS framework but does not hold for all models. For instance, diffusion models are trained via a score-matching objective across many noise levels, and their sample quality is not directly coupled with the fidelity of a single-step reconstruction in the way an autoencoder's is. Moreover, the assertion that the overlap coefficient for GANs is effectively 1 is not justified.
3. The theoretical results, Theorems 3.3 and 3.5, appear to be formalizations of rather intuitive and well-understood principles. For example, Theorem 3.3 states that reconstruction error is lower-bounded by the overlap of latent distributions. This is almost a definitional consequence: if a region of the latent space is mapped to multiple distinct targets, any deterministic decoder will necessarily incur an error. The theorem formalizes this intuition but does not provide a particularly deep or surprising insight into the learning process.

**Questions:**

See above.

---

### Official Review · Reviewer_676g · 2025-11-01

**Soundness:** 2
**Presentation:** 1
**Contribution:** 2
**Rating:** 2
**Confidence:** 3

**Summary:**

This paper challenges the assumption in generative modeling that learning a global data distribution is sufficient for novel synthesis. The authors argue that this approach instead leads to memorization. To investigate this, they introduce two theoretical frameworks: the Mutually Exclusive Probability Space (MEPS), which suggests that models naturally separate latent distributions to improve fidelity, and the Local Dependence Hypothesis (LDH), which posits that true generative capacity stems from modeling local, not global, dependencies. As a practical application of MEPS, the authors propose a Binary Latent Autoencoder (BL-AE) to encode images into signed binary representations. To test LDH, they introduce the gamma-Autoregressive Random Variable Model (gamma-ARVM), an autoregressive model with a variable context window. Through experiments, they demonstrate that as the observation range increases, the model progressively shifts from generative behavior to memorization, culminating in near-perfect reproduction of training samples when operating on the binary latents with a global context.

**Strengths:**

- The paper's discussion on memorization, generation, and reconstruction is interesting.
- The numerical results on experiments of reconstruction are impressive.

**Weaknesses:**

- The paper's conclusion appears to be in contradiction with the recent advancements of autoregressive model LLMs, which lie precisely in its powerful ability to model long-range dependencies. The author's findings that increasing the observation range leads to memorization is at odds with this trend. The author needs to discuss it in the paper.
- The paper's core argument is only tested on an autoregressive model but is unfairly generalized to all generative models, failing to explain how it applies to high-performing architectures like Diffusion Models, which also leverage global information.
- The paper needs to differentiate the proposed BL-AE from existing binary encoding methods like LFQ to justify its novelty.
- The generation experiments, conducted on only 1k images, are too small to draw meaningful conclusions about memorization versus overfitting. The findings require validation on larger-scale datasets.

**Questions:**

See weakness

---

### Note · Authors · 2025-12-12

I have read and agree with the venue's withdrawal policy on behalf of myself and my co-authors.